# Effect of one dose of ceftriaxone during endotracheal intubation on the incidence of aspiration pneumonia in cerebral hemorrhage patients: A randomized, controlled, double-blind clinical study protocol

**Xinyan Liu[1]ʘ, Qizhi Wang[2]ʘ, Yang Bi[3], Yanru Yue[3], Xuan Song**ᴑᴿᶜᴵᴰ[2,4]*

**1** ICU, Dong E Hospital, Liaocheng, Shandong Province, China, **2** ICU, Shandong Provincial Hospital affiliated to Shandong First Medical University, Jinan, Shandong Province, China, **3** Shandong First Medical University, Jinan, Shandong Province, China, **4** Shandong Institute of Endocrine & Metabolic Diseases affiliated to Shandong First Medical University, Jinan, Shandong, China

ʘ These authors contributed equally to this work.
* songxuan0303@163.com

## Abstract

### Introduction

Patients with cerebral hemorrhage often require a tracheal intubation to protect the airway and maintain oxygenation. Due to the use of analgesic and sedative drugs during endotracheal intubation and the opening of the glottis may easily cause aspiration pneumonia. Ceftriaxone is a semi-synthetic third-generation cephalosporin with strong antimicrobial activity against most gram-positive and gram-negative bacteria. It can effectively prevent and treat aspiration pneumonia.

### Methods and analysis

This is a prospective, randomized, controlled, double-blind clinical study. Patients with intracerebral hemorrhage (ICH) undergoing endotracheal intubation in Dong E Hospital of Shandong Province from April 2023 to April 2025 will be enrolled and randomly assigned to the intervention group or control group. The intervention group will be treated using 100mL 0.9% sodium chloride with 2g ceftriaxone intravenously over the course of one hour beginning within two hours after endotracheal intubation. The control group will be given 100mL 0.9% sodium chloride injection intravenously of the course of one hour beginning within two hours after endotracheal intubation. The primary outcome is the incidence of aspiration pneumonia within 48 hours after endotracheal intubation. Secondary outcomes include: intensity of antimicrobial use, length of hospital stay, duration without mechanical ventilation, and 28-day mortality.

**Data Availability Statement:** Deidentified research data will be made publicly available when the study is completed and published.

**Funding:** This work was supported by Natural Science Foundation of Shandong Province, China (XS, Grant ZR202102210272), Taishan Scholars Program of Shandong Province (XS, Grant tsqn202211347) and Shandong Provincial Postdoctoral Science Foundation (XS, Grant SDCX-ZG-202202025). The funders did not and will not have a role in study design, data collection and analysis, decision to publish, or preparation of the manuscript.

**Competing interests:** The authors have declared that no competing interests exist.

## Discussion

The primary objective of this study is to explore whether a single dose of ceftriaxone administered during endotracheal intubation in patients with ICH reduced the incidence of pneumonia within 48 hours and provide evidence for the prevention of aspiration pneumonia in patients with ICH with endotracheal intubation.

## Trial registration

The trial is registered at the Chinese Clinical Trial Registry: ChiCTR2200066837. Registered on December 19, 2022.

## Introduction

Patients with intracerebral hemorrhage (ICH) often experience such complications as consciousness disorders, respiratory center damage, airway obstruction, and swallowing dysfunction [1]. Endotracheal intubation is often required to protect the airway and maintain oxygenation [2]. However, during intubation, analgesic and sedative drugs are administered and the glottis is exposed, allowing oral secretions to easily to enter the lower respiratory tract and cause aspiration pneumonia [3–5]. Inhaled substances may include secretions from the nasal cavity and oropharynx, residual food in the mouth, contents of gastrointestinal vomiting, and digestive fluid reflux, which may cause aspiration pneumonia [6]. One type of aspiration pneumonia is chemical inflammation caused by direct damage to lung tissue caused by inhalation of gastric contents; another type is pulmonary atelectasis and inflammation caused by inhalation of obstructive substances; and the third type is bacterial pneumonia caused by inhalation of oral secretions containing colonized bacteria, which is the most common clinical type and can be fatal in severe cases [7]. Studies have shown that the expected mortality of patients with aspiration pneumonia is higher than that of patients with other forms of pneumonia [8].

Pneumonia caused by inhalation of anaerobic bacteria may be subacute because of the low virulence of the bacteria, but its clinical features are difficult to distinguish from other bacterial pneumonitis [9]. While both aspiration pneumonia and aspiration pneumonitis result from the inhalation of foreign substances into the lungs, they differ in terms of the infectious nature of the lung injury. Aspiration pneumonia involves a bacterial infection following an aspiration event, whereas aspiration pneumonitis is characterized by an inflammatory reaction to irritative gastric contents without the development of bacterial infection. Clinicians should be aware of the differences between these conditions to ensure appropriate management and treatment.

A prospective observational study involving a total of 879 cases of emergency endotracheal intubation patients found that, after excluding patients who were not eligible for prognostic assessment (those who died within 48 hours without detection of pneumonia), 8.0% (66/823) developed aspiration pneumonia that may have been associated with emergency intubation [10]. A large study from a single center in Japan showed that the incidence of aspiration pneumonia in elderly patients (aged >70) was 67.2 per 10,000 people from 2010 to 2015, and the mortality rate was 13.6% [11]. A retrospective study including patients with an average age of 78 in a single center in South Korea showed that the 1-year mortality rate of patients with aspiration pneumonia was 49.0% [12]. Even when the data are corrected, the case fatality rate of

aspiration pneumonia is still higher than that of other forms of pneumonia [12]. Therefore, the prevention of aspiration pneumonia is particularly important.

Conventional measures to prevent aspiration pneumonia include: elevation of the head of the bed, good oral care, swallowing function assessment and training, comprehensive rehabilitation, nasogastric feeding, improvement of gastric emptying function, and monitoring of gastric residual volume [13]. Given the pathophysiology of aspiration pneumonia and infection caused by aspiration, the application of antibiotics before endotracheal intubation can be regarded as a preventive strategy. In the *Expert Advice on the Diagnosis and Treatment of Aspiration Pneumonia in Adults*, the preventive measures for aspiration pneumonia are recommended as 'antibiotic treatment for 24 hours after emergency intubation in coma patients' [14].

Previous studies have shown that anaerobic bacterial infections are not common in aspiration pneumonia, with pathogens like *enterobacteria*, *Haemophilus influenzae*, *Streptococcus pneumoniae*, and *Staphylococcus aureus* being more prevalent in community-acquired cases, and gram-negative bacilli, such as *Pseudomonas aeruginosa*, in hospital-acquired cases [15]. Only one case of ventilator associated pneumonia was the result of an anaerobic infection [16]. In cases of post-stroke pneumonia, gram-negative bacilli (mainly *Klebsiella pneumoniae* and *Escherichia coli*) and *Staphylococcus aureus* were frequently isolated [17]. Similarly, in patients with intracerebral hemorrhage (ICH) who underwent endotracheal intubation, nosocomial respiratory infections were found in over half of the cases, with gram-negative bacteria being the predominant pathogens [18]. Ceftriaxone, a semi-synthetic third-generation cephalosporin, is highly effective against these common pathogens, making it a suitable option for preventing and treating aspiration pneumonia in these patients [19, 20].

The aim of this study is to investigate whether a single dose of ceftriaxone during endotracheal intubation can reduce the incidence of pneumonia within 48 hours in patients with cerebral hemorrhage, and to provide a basis for the prevention of aspiration pneumonia in patients with cerebral hemorrhage.

## Methods and analysis

This study was approved by the hospital ethics committees of Dong E Hospital. All methods will be performed in accordance with the relevant guidelines and regulations. Written informed consent will be obtained from the guardians of all patients and registered in the Chinese Clinical Trial Registry (Registration number: ChiCTR2200066837). All methods were reported according to Consolidated Standards of Reporting Trials (CONSORT) [21].

### Study design

This is a prospective, randomized, controlled, double-blind clinical study, which was reviewed by the ethics committee of Dong E Hospital.

### Inclusion and exclusion criteria

Patients with ICH undergoing endotracheal intubation in Dong E Hospital of Shandong Province from April 2023 to April 2024 will be included. Inclusion criteria are: (1) age > 18 years; (2) ICH due to various causes; (3) endotracheal intubation; and (4) subjects understood and signed the informed consent. The following patients will be excluded from this study: (1) pneumonia occurred before endotracheal intubation; (2) aspiration occurred before endotracheal intubation (e.g., aspiration that is considered possible clinically, such as epilepsy; prehospital vomiting; vomit on the patient's clothing; or vomit visible in the oropharynx at the beginning of intubation attempts); (3) received antibiotics before endotracheal intubation; (4) received other antibiotics within 48 hours after enrollment (especially before diagnosis of

| | STUDY PERIOD | | | | | | |
|---|---|---|---|---|---|---|---|
| | Enrolment | Allocation | Post-allocation | | | | Close-out |
| **TIMEPOINT** | $-t_1$ | **0** | $t_1$ | $t_2$ | $t_3$ | $t_4$ | $t_x$ |
| **ENROLMENT:** | | | | | | | |
| **Eligibility screen** | X | | | | | | |
| **Informed consent** | X | | | | | | |
| *[List other procedures]* | X | | | | | | |
| **Allocation** | | X | | | | | |
| **INTERVENTIONS:** | | | | | | | |
| *[Intervention A]* | | | X | | | | |
| *[Intervention B]* | | | | | | | |
| *[List other study groups]* | | | ●━━━━━━━━● | | | | |
| **ASSESSMENTS:** | | | | | | | |
| *[List baseline variables]* | X | X | | | | | |
| *[List outcome variables]* | | | | X | | X. | X |
| *[List other data variables]* | | | X | X | X | X. | X |

**Fig 1. Schedule of enrolment, interventions, and assessments.**

aspiration pneumonia); (5) patients who died within 48 hours after endotracheal intubation without aspiration pneumonia; (6) allergic history to cephalosporin antibiotics; and (7) drinking history within one week before endotracheal intubation, as the use of cephalosporin after drinking alcohol can cause a disulfiram reaction. Of note, patients with COPD will not be excluded from the study. Schedule of enrolment, interventions, and assessments are shown in **Fig 1**. A simplified schematic of the trial design is shown in **Fig 2**.

## Intervention

Subjects will be randomly assigned 1:1 to the ceftriaxone intervention group or the control group after signing informed consent on a computer-generated randomization table generated by an investigator not involved in the clinical trial. Patients who meet the inclusion criteria will undergo endotracheal intubation routinely according to the operating specifications of

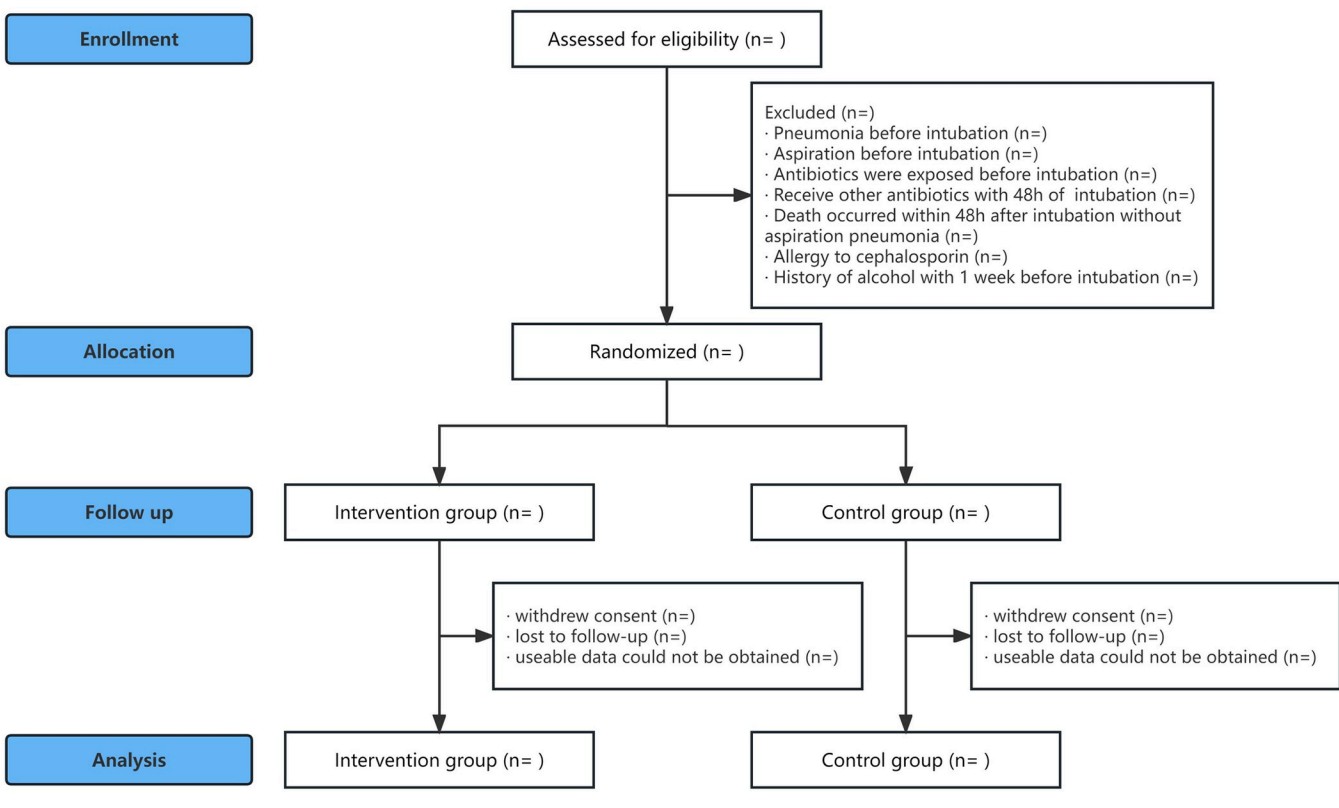

**Fig 2. Simplified schematic of the trial design.**

our unit. Patients in the intervention group will be given 100mL 0.9% sodium chloride with 2g ceftriaxone sodium intravenously over the course of one hour within two hours after endotracheal intubation. Patients in the control group will receive 100 mL 0.9% sodium chloride injection intravenously over the course of one hour within two hours after endotracheal intubation. Other treatment measures after endotracheal intubation (ie: prevention of aspiration, withdrawal of mechanical ventilation) are the same as conventional treatment. Neither the investigators nor the patients will be aware of the group assignments during the study. Open label is allowed in case of serious adverse effects such as allergic reaction.

## Data collection

**Patient characteristics.** The general data of the patients will be collected, including gender, age, height, weight, BMI, comorbidities, Acute Physiology and Chronic Health Evaluation II (APACHE II) score, Sequential Organ Failure Assessment (SOFA) score, history of medication allergy, diagnosis, smoking history, drinking history, bed rest history, chest radiograph or chest CT manifestations before endotracheal intubation, laboratory test results (white blood cell count, neutrophil ratio, C-reactive protein and procalcitonin), body temperature, cause of ICH, time of onset of aspiration pneumonia after enrollment, duration of mechanical ventilation, length of ICU stay, length of hospital stay, and 28-day survival.

**Intubation characteristics.** Intubation characteristics will be collected, including whether vomiting occurred before endotracheal intubation; whether obvious aspiration occurred before endotracheal intubation; whether aspiration occurred during endotracheal intubation; whether mechanically ventilated; endotracheal intubation time; endotracheal intubation type;

whether analgesic and sedative drugs were used before intubation; whether intubation was difficult; whether repeated intubation attempts were necessary.

**Antibiotic characteristics.** Antibiotic characteristics will be collected, including time of administration, antibiotic type, dose, frequency, and days of antibiotic use after enrollment; sputum or broncho alveolar lavage fluid (BALF) culture results during endotracheal intubation and sputum or BALF culture results during aspiration pneumonia.

## Research outcomes

The primary outcome is the incidence of aspiration pneumonia within 48 hours after endotracheal intubation. Aspiration pneumonia is defined the presence of new or advanced infiltrating shadows, consolidation shadow, or ground glass shadow on chest X-ray or CT in patients without a diagnosis of community-acquired pneumonia, healthcare-associated pneumonia, or aspiration pneumonia, combined with two or more of the following clinical symptoms: (1) fever, body temperature >38˚C; (2) purulent airway discharge; (3) white blood cell count >10 $\times10 \wedge 9$ / L or<4×10^9/L. Secondary outcomes include: intensity of antimicrobial use, length of hospital stay, duration without mechanical ventilation, and 28-day mortality. A subgroup analysis will be conducted in patients with high and low aspiration pneumonia risk, which will be defined as follows: low-risk patients will exhibit none of the following intubation characteristics: visible stomach contents during intubation, aspiration suspected, and evidence of aspiration by bronchoscopy. High risk patients will exhibit one or more of the following intubation characteristics: visible stomach contents during intubation, aspiration suspected, and evidence of aspiration by bronchoscopy.

The intensity of antimicrobial use is defined as:

$$\frac{\text{Antimicrobial consumption (cumulative DDD)}}{\text{Days of patients admitted in the same period}} \times 100$$ where DDD is the defined daily dose.

The intensity of antibiotic use in a particular patient is defined as:

$$\frac{\text{Antimicrobial consumption (cumulative DDD)}}{\text{Days of hospital stay}} \times 100$$

$$\frac{\text{Antimicrobial drug consumption}/DDD}{\text{Days of hospital stay}} \times 100$$

$$\frac{\text{Single dose of antibiotics} \times \text{frequency of use} \times \text{days of use}/DDD}{\text{Days of hospital stay}} \times 100$$

## Adverse event monitoring

Patients with a history of allergy to cephalosporins will be excluded to avoid allergic reactions to the drugs. Allergic reactions include high fever, rash, pruritus, dyspnea, laryngeal edema, and anaphylactic shock. Once allergy occurs, the drug will be stopped immediately, and epinephrine or appropriate first aid measures (e.g., airway management, intravenous rehydration, antihistamines, corticosteroids, vasoactive drugs) will be given to patients with severe acute hypersensitivity reaction according to clinical indications. Serious adverse reactions will be reported to the Ethics Committee immediately.

## Sample size calculation

A total of 50 patients will be included in the preliminary study, with 25 patients in each of the ceftriaxone and control groups. For the preliminary results, the incidence of pneumonia within 48 hours after endotracheal intubation in the two groups was 16%(4/25) and 32%(8/25), respectively. The sample size estimation formula for the comparison of two sample rates will be used to estimate the sample size; the test level α was set at 0.05, and the type Ⅱ error

probability β was set at 0.2. The patients will be included according to 1:1 randomization, and the required sample size of each group was calculated to be 177 cases by two-sided test. Considering the loss of follow-up, based on the loss rate of 10%, at least 197 cases subjects were needed in the each group.

## Statistical analysis

SPSS 20.0 will be used for statistical analysis. Demographic factors and baseline characteristics, such as age, sex, APACHE II score, and SOFA score, will be adjusted for using appropriate statistical methods. These adjustments may include logistic regression for binary outcomes or stratified analysis if needed. The primary outcome, which is the incidence of aspiration pneumonia within 48 hours of intubation, will be compared between the control and intervention groups using either the chi-square test or Fisher's exact test, depending on the distribution of the categorical data. For the secondary outcomes, continuous variables like length of ICU stay, duration of mechanical ventilation, and hospital length of stay will be described using mean ± standard deviation or median (interquartile range) depending on the distribution of the data. Differences between the groups will be tested using independent t-tests for normally distributed data or Mann-Whitney U tests for non-normally distributed data. Categorical outcomes such as 28-day mortality will be analyzed using chi-square or Fisher's exact test. For any time-to-event outcomes, such as the time until the development of aspiration pneumonia or extubation, Kaplan-Meier survival analysis will be used, with differences between the groups assessed via the log-rank test.

In terms of assumptions, the study will check for normality and independence in the data, particularly for tests like the t-test or chi-square test. If assumptions of normality are violated, appropriate data transformations (e.g., logarithmic transformation) or non-parametric tests will be used. For categorical data, Fisher's exact test will be used if expected cell counts are less than 5 in any of the categories. Missing data will be handled using a complete case analysis, where possible. If missing data is substantial, multiple imputation techniques will be applied to ensure the integrity and reliability of the analysis.

## Discussion

This is a prospective, randomized, controlled, double-blind clinical study to compare the effect of a single dose of ceftriaxone during endotracheal intubation on the incidence of aspiration pneumonia within 48 hours after endotracheal intubation in patients with ICH. This will be the first clinical study of ceftriaxone to prevent the development of aspiration pneumonia in patients with ICH hemorrhage undergoing endotracheal intubation, closing a critical gap in the literature.

A recent study on oral microecology found 103 species of bacteria in the mouth of patients with acute stroke; 29 of these were previously unreported bacteria of which the pathogenicity is still unknown [22]. Previous studies have shown that aspiration pneumonia is often due to infection with gram-negative bacteria (e.g., *E. coli*, *K. pneumoniae*, *P. aeruginosa*) and *S. aureus* [7, 14], and antibiotics used prophylactically should target these pathogens. Ceftriaxone, which is effective against both gram-positive and gram-negative bacteria, was found to be an effective agent for the prevention of bacterial aspiration pneumonia in intubated ICH patients. For this reason, we selected ceftriaxone as the treatment in this study.

In a study evaluating the risk factors for pneumonia within 48 hours after intubation, Rello et al. [23] found that patients with respiratory/cardiac arrest and coma had the highest incidence of pneumonia in the first 48 hours. Antibiotic use during this period reduced the incidence of pneumonia, while other preventive measures such as subglottic suction were

ineffective. Sirvent et al. [24] conducted a randomized trial to evaluate the effectiveness of two doses of cefuroxime administered within 24 hours after intubation in reducing the incidence of early-onset ventilator-associated pneumonia (VAP) in patients with closed head injury (Glasgow Coma Scale score < 12). A retrospective cohort study by Vlad Dragan et al. [25] reached a different conclusion, finding instead that prophylactic antimicrobial therapy in patients with acute aspiration pneumonia does not provide clinical benefit, has no effect on mortality improvement, and may generate antibiotic selection pressure and increase the number of days of subsequent antibiotic treatment. In these cases, however, antibiotic use may be more akin to short-term treatment than true prevention, and long-term antibiotic use may increase the risk of subsequent infection by resistant microorganisms. For this reason, we decided to use a single dose of ceftriaxone, an antibiotic which has a long half-life and is effective against the main pathogens responsible for most endotracheal intubation-induced aspiration pneumonia.

A previous similar study by Valles et al. [26] showed that a single dose of antibiotics significantly reduced the incidence of VAP in mechanically ventilated comatose patients. It also reduced mechanical ventilation and ICU length of stay, and had no effect on mortality or the incidence of multidrug-resistant microbial infections. However, this was a non-randomized study, and there was bias in the selection of patients in the control group. The incidence of chronic obstructive pulmonary disease in the control group was higher, which resulted in a greater degree of airway pathogen colonization before intubation, and more patients requiring neurosurgical intervention were included in the control group. Our study will be randomized, making it a more effective analysis of a single antibiotic dose as a prophylactic measure for aspiration pneumonia.

The use of prophylactic antibiotics in aspiration pneumonia presents a complex balance between potential benefits and risks. While some studies suggest a potential benefit in reducing the risk of acquiring aspiration pneumonia [27], others indicate limited efficacy and potential adverse effects, such as the escalation of antibiotic therapy and the promotion of drug-resistant bacteria [28]. Prophylactic antibiotic therapy has been associated with a lower risk of acquiring aspiration pneumonia in patients with dysphagia [27], but conflicting evidence exists regarding its efficacy in reducing the incidence of pneumonia in patients with acute stroke and dysphagia [29]. Additionally, prophylactic antibiotic use may not prevent stroke-associated pneumonia and could potentially increase the risk of pneumonia [30]. Therefore, the decision to use prophylactic antibiotics in aspiration pneumonia should be carefully evaluated, considering the individual patient's risk factors and the potential for adverse effects.

Further prospective randomized controlled studies are necessary to explore whether a dose of ceftriaxone before endotracheal intubation can reduce the incidence of aspiration pneumonia in patients with ICH, to provide an option for the prevention of pneumonia in these patients.

Our study has some limitations. First, this study will be conducted in a single center and will include a small sample size, which may limit the broader applicability of our findings. Second, the definition of aspiration pneumonia may inadvertently include aspiration pneumonia before endotracheal intubation because we will exclude these patients on the basis of clinical speculation. Finally, it is difficult to distinguish between aspiration pneumonia and another type of pneumonia that develops within 48 hours of endotracheal intubation.

## Trial status

At the time of manuscript submission, the study is in the preparation phase for recruitment. This is the first version of the protocol completed on September 30, 2022. Recruitment is scheduled to begin in April 2023 and expected to be completed by April 2025.

## Author Contributions

**Conceptualization:** Xuan Song.

**Data curation:** Yanru Yue.

**Formal analysis:** Yanru Yue.

**Investigation:** Xinyan Liu, Qizhi Wang.

**Methodology:** Xinyan Liu.

**Project administration:** Xuan Song.

**Software:** Yang Bi.

**Supervision:** Xuan Song.

**Validation:** Xinyan Liu, Yang Bi.

**Visualization:** Xinyan Liu, Qizhi Wang.

**Writing – original draft:** Xinyan Liu, Qizhi Wang.

**Writing – review & editing:** Xuan Song.

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
