## [Decision Letter · Decision Letter 0]

17 Sep 2023

PONE-D-23-05214Effect of one dose of ceftriaxone during endotracheal intubation on the incidence of aspiration pneumonia in cerebral hemorrhage patients: A randomized, controlled, double-blind clinical study protocolPLOS ONE

Dear Dr. Song,

Thank you for submitting your manuscript to PLOS ONE. After careful consideration, we feel that it has merit but does not fully meet PLOS ONE’s publication criteria as it currently stands. Therefore, we invite you to submit a revised version of the manuscript that addresses the points raised during the review process.

We look forward to receiving your revised manuscript.

Kind regards,

Burak Katipoğlu

Academic Editor

PLOS ONE

Journal Requirements:

Additional Editor Comments :

Dear Authors, Thank you for giving us the opportunity to review this article.

I have some concern for this manuscript.

Could you explain why you chose the cerebral hemorrhage group?

In the discussion section, write down the differences and similarities from the articles in the literature, together with the reasons.

Kind Regards.

Reviewers' comments:

Reviewer's Responses to Questions

**Comments to the Author**

1. Does the manuscript provide a valid rationale for the proposed study, with clearly identified and justified research questions?

Reviewer #1: Yes

Reviewer #2: Yes

Reviewer #3: Partly

2. Is the protocol technically sound and planned in a manner that will lead to a meaningful outcome and allow testing the stated hypotheses?

Reviewer #1: Yes

Reviewer #2: Yes

Reviewer #3: Partly

3. Is the methodology feasible and described in sufficient detail to allow the work to be replicable?

Reviewer #1: Yes

Reviewer #2: Yes

Reviewer #3: No

4. Have the authors described where all data underlying the findings will be made available when the study is complete?

Reviewer #1: Yes

Reviewer #2: Yes

Reviewer #3: No

5. Is the manuscript presented in an intelligible fashion and written in standard English?

Reviewer #1: Yes

Reviewer #2: Yes

Reviewer #3: Yes

6. Review Comments to the Author

You may also provide optional suggestions and comments to authors that they might find helpful in planning their study.

Reviewer #1: I thank the editor for submitting the work of the Chinese group for review.

This work consists of a double-blind randomized controlled trial protocol that seeks to evaluate whether it is possible to decrease the incidence, with a single dose of cephalosporin, of aspiration pneumonia in a specific test population, those subject to intracerebral hemorrhage.

Keywords.

1. I congratulate the authors for their choice of keywords. All relevant and found in the reference database.

Introduction

2. To check reference #1, as wording or citation.

3. Line 66-74 I advise the authors, in this part of the introduction to include some reference to the sentences said, or, if they are speculations or statements of the authors perhaps rephrase the sentences.

4. Line 76-85 I advise the authors in this passage to specify the type of patients examined by the studies mentioned, for the study in reference no. 2 ALL emergency intubated patients were taken, for studies instead 3 and 4 patients were examined, for study 3 elderly, after the age of 70 years stated, and for study 4 outcome of a population with a median age of 78 years is evaluated.

This is a point to be specified in the introduction for clarity and accuracy.

5. Line 116-121 To be included references to this. They are not in the document.

Study design

6. I would like to ask the authors how they plan to deal with COPD patients, patients who are more delicate and who may also be more prone to pneumonia, both nosocomial but especially community-acquired pneumonia. Will they be excluded or included in the study and then perhaps analyzed separately? This is a subcategory to be carefully evaluated.

Figure 1 and 2

7. Regarding the figures I would have some clarifications for the authors to make. Figure 1 is interesting in that it describes, adequately, the pattern of enrollment, interventions, and assessments done.

8. For Figure 2, it would also be to complete after the intervention and control boxes, with perhaps the patients lost and the amount of patients analyzed.

9. Regarding exclusion criterion 7, however, I would specify the reason for this criterion. Why were those who had a history of alcohol consumption in the week prior to endotracheal intubation not considered? For what reason? Is there a reference that discourages this type of patient in the possibility of being enrolled?

Intervention

10. Line 162-163 Why was the infusion rate of ceftriaxone chosen to be 1 hour and not 30 minutes given that ceftriaxone is known to be a time-dependent drug and acts on the plasma peak and not on maintaining a constant dose? This is a point that needs to be evaluated and clarified in order to be timely in describing the intervention, which is also a cornerstone on which the study is based, the administration of the antibiotic.

11. Line 168. I advise authors to state serious adverse events and not take them for granted, I assume symptoms of severe allergic crisis in patients not known allergic, but to be stated.

Data collection

12. I would advise the authors to subdivide this subsection further, perhaps putting, patient characteristics, intubation characteristics, and antibiotic therapy. So that the data collected can be visualized better and more schematically.

13. Additional data that I would also collect are laboratory tests at admission, at least those related to infectious status, considering that later we are going to evaluate an infection, such as (Body temperature, complete hemogram with complete Leukocyte formula, Thrombus and CRP and procalcitonin, and maybe other values if present and measured), so that we have a start and know possibly whether to exclude from the study patients who do not have obvious fever but maybe had an initial infectious status. And possibly even if there is a possibility, of a bronchial secretion analysis at the time of intubation, or even better a BAL.

Statistical analysis

14. I recommend that the authors also state that the data will be described statistically with mean and standard deviation or median and quartile depending on the distribution

Discussion

15. Line 276-278 Appropriate reference to the authors' statement should be included.

Reviewer #2: 1) There is a mistake in the date in methods and analysis (line 41) :it should be from April 2023 to April 2024.

2) Please provide a copy of ethical committee approval for the trial.

Reviewer #3: Dear Authors, Thank you for giving us the opportunity to review this article titled “Effect of one dose of ceftriaxone during endotracheal intubation on the incidence of aspiration pneumonia in cerebral hemorrhage patients: A randomized, controlled, double-blind clinical study protocol”.

I don't have any conflict of interest.

I have some concern for this manuscript.

First of all, this should be clarified. Should antibiotics be given before or after intubation? In some sentences, antibiotics are mentioned before intubation and in others after.

The introduction part is too long.

bacteria should be given briefly.

exclusion criteria

consciousness changes?

Actually, isn't this the riskiest group?

Therefore, informed consent should be obtained from their relatives. I think excluding the most risky patient group will impoverish this study.

History of drinking in the week before endotracheal intubation. (why? reference?)

Why is ceftriaxone given as an infusion for 1 hour?

Will each patient be paag after intubation? to exclude old infiltrates?

data collection

Has a mechanical application been made or not to prevent aspiration?

Will this information be recorded?

The discussion part should be completely rewritten.

And, The most important question that needs to be explained is this.

Why cerebral hemorrhage patient group?

Kind Regards.

7. PLOS authors have the option to publish the peer review history of their article (what does this mean?). If published, this will include your full peer review and any attached files.

Reviewer #1: No

Reviewer #2: No

Reviewer #3: No

---

## [Author Response · Author response to Decision Letter 0]

16 Oct 2023

Dear PLOS One Editors,

Thank you for your thoughtful comments and suggestions for our manuscript, “Effect of one dose of ceftriaxone during endotracheal intubation on the incidence of aspiration pneumonia in cerebral hemorrhage patients: A randomized, controlled, double-blind clinical study protocol”. Below in blue are responses to Editor and Reviewer comments that you will also find addressed in our manuscript. We appreciate the opportunity to address the comments and resubmit an improved manuscript for considered publication in PLOS One.

Sincerely,

Xuan Song, MD, PhD

Comments to the Author

Reviewer #1: I thank the editor for submitting the work of the Chinese group for review.

This work consists of a double-blind randomized controlled trial protocol that seeks to evaluate whether it is possible to decrease the incidence, with a single dose of cephalosporin, of aspiration pneumonia in a specific test population, those subject to intracerebral hemorrhage.

Keywords.

1.I congratulate the authors for their choice of keywords. All relevant and found in the reference database.

Thank you.

Introduction

2.To check reference #1, as wording or citation.

This has been addressed.

3.Line 66-74 I advise the authors, in this part of the introduction to include some reference to the sentences said, or, if they are speculations or statements of the authors perhaps rephrase the sentences.

We have added references to these sentences.

4. Line 76-85 I advise the authors in this passage to specify the type of patients examined by the studies mentioned, for the study in reference no. 2 ALL emergency intubated patients were taken, for studies instead 3 and 4 patients were examined, for study 3 elderly, after the age of 70 years stated, and for study 4 outcome of a population with a median age of 78 years is evaluated.

This is a point to be specified in the introduction for clarity and accuracy.

We have specified the types of patients in the above studies in the introduction.

4.Line 116-121 To be included references to this. They are not in the document.

We have added references to these sentences.

Study design

6.I would like to ask the authors how they plan to deal with COPD patients, patients who are more delicate and who may also be more prone to pneumonia, both nosocomial but especially community-acquired pneumonia. Will they be excluded or included in the study and then perhaps analyzed separately? This is a subcategory to be carefully evaluated.

These patients are not excluded, but will be compared in baseline characteristics. We have noted this in the manuscript. 

Figure 1 and 2

7.Regarding the figures I would have some clarifications for the authors to make. Figure 1 is interesting in that it describes, adequately, the pattern of enrollment, interventions, and assessments done.

Thank you.

8. For Figure 2, it would also be to complete after the intervention and control boxes, with perhaps the patients lost and the amount of patients analyzed.

Thank you for your suggestion. We have revised Figure 2 according to the steps of the randomized controlled trial.

9. Regarding exclusion criterion 7, however, I would specify the reason for this criterion. Why were those who had a history of alcohol consumption in the week prior to endotracheal intubation not considered? For what reason? Is there a reference that discourages this type of patient in the possibility of being enrolled?

Thank you for your question. The use of cephalosporin after drinking alcohol can easily cause disulfiram reaction.The manifestation of CIDLR (cephalosporin-induced disulfiram-like reaction) varies from mild reaction like facial flushing, nausea or vomiting, to a severe reaction including angioedema, hypotension, shock, or death (1,2). Therefore, we excluded such patients in order to avoid risk.

(1)Mergenhagen KA, Wattengel BA, Skelly MK, Clark CM, Russo TA. Fact versus Fiction: a Review of the Evidence behind Alcohol and Antibiotic Interactions. Antimicrob Agents Chemother. 2020;64(3):e02167-19. Published 2020 Feb 21. doi:10.1128/AAC.02167-19

(2) Ren S, Cao Y, Zhang X, Jiao S, Qian S, Liu P. Cephalosporin induced disulfiram-like reaction: a retrospective review of 78 cases. Int Surg. 2014;99(2):142-146. doi:10.9738/INTSURG-D-13-00086.1

Intervention

10. Line 162-163 Why was the infusion rate of ceftriaxone chosen to be 1 hour and not 30 minutes given that ceftriaxone is known to be a time-dependent drug and acts on the plasma peak and not on maintaining a constant dose? This is a point that needs to be evaluated and clarified in order to be timely in describing the intervention, which is also a cornerstone on which the study is based, the administration of the antibiotic.

Thank you for your question. According to the drug instructions of ceftriaxone sodium, the infusion time should be at least 30min to avoid adverse reactions, so we defined the infusion time as 1 hour in the protocol.

11. Line 168. I advise authors to state serious adverse events and not take them for granted, I assume symptoms of severe allergic crisis in patients not known allergic, but to be stated.

We now include this information in the manuscript.

Data collection

12.I would advise the authors to subdivide this subsection further, perhaps putting, patient characteristics, intubation characteristics, and antibiotic therapy. So that the data collected can be visualized better and more schematically.

Agree with the reviewer. We have subdivided Data Collection into the above sections.

13. Additional data that I would also collect are laboratory tests at admission, at least those related to infectious status, considering that later we are going to evaluate an infection, such as (Body temperature, complete hemogram with complete Leukocyte formula, Thrombus and CRP and procalcitonin, and maybe other values if present and measured), so that we have a start and know possibly whether to exclude from the study patients who do not have obvious fever but maybe had an initial infectious status. And possibly even if there is a possibility, of a bronchial secretion analysis at the time of intubation, or even better a BAL.

Agree with the reviewer. We have included laboratory test results in the information to be collected.

Statistical analysis

14. I recommend that the authors also state that the data will be described statistically with mean and standard deviation or median and quartile depending on the distribution.

Agree with the reviewer. We have included this information in the manuscript.

Discussion

15. Line 276-278 Appropriate reference to the authors' statement should be included.

Reviewer #2: 1) There is a mistake in the date in methods and analysis (line 41) :it should be from April 2023 to April 2024.

Thank you. We have corrected this mistake.

2) Please provide a copy of ethical committee approval for the trial.

Ok, we submitted the copy of ethical committee approval of the trial.

Reviewer #3: Dear Authors, Thank you for giving us the opportunity to review this article titled “Effect of one dose of ceftriaxone during endotracheal intubation on the incidence of aspiration pneumonia in cerebral hemorrhage patients: A randomized, controlled, double-blind clinical study protocol”.

I don't have any conflict of interest.

I have some concern for this manuscript.

First of all, this should be clarified. Should antibiotics be given before or after intubation? In some sentences, antibiotics are mentioned before intubation and in others after.

Due to the urgency of rescuing patients in clinical practice, we cannot guarantee the use of ceftriaxone before tracheal intubation. According to our protocol, it must be used within 2 hours after tracheal intubation.

The introduction part is too long.

We have endeavored to shorten the introduction.

bacteria should be given briefly.

We have endeavored to shorten the bacteria discussion in the introduction, but feel it is important to note the kinds of bacteria that are found in aspiration pneumonia cases.

exclusion criteria

consciousness changes?

Actually, isn't this the riskiest group?

Therefore, informed consent should be obtained from their relatives. I think excluding the most risky patient group will impoverish this study.

I can't agree with you more. We included patients with consciousness changes in our protocol and primarily excluded those who had an impact on aspiration and aspiration pneumonia. We seek informed consent from their relatives with consciousness changes.

History of drinking in the week before endotracheal intubation. (why? reference?)

Thank you for your question. The use of cephalosporin after drinking alcohol can easily cause disulfiram reaction.The manifestation of CIDLR (cephalosporin-induced disulfiram-like reaction) varies from mild reaction like facial flushing, nausea or vomiting, to a severe reaction including angioedema, hypotension, shock, or death (1,2). Therefore, we excluded such patients in order to avoid risk.

(1)Mergenhagen KA, Wattengel BA, Skelly MK, Clark CM, Russo TA. Fact versus Fiction: a Review of the Evidence behind Alcohol and Antibiotic Interactions. Antimicrob Agents Chemother. 2020;64(3):e02167-19. Published 2020 Feb 21. doi:10.1128/AAC.02167-19

(2) Ren S, Cao Y, Zhang X, Jiao S, Qian S, Liu P. Cephalosporin induced disulfiram-like reaction: a retrospective review of 78 cases. Int Surg. 2014;99(2):142-146. doi:10.9738/INTSURG-D-13-00086.1

Why is ceftriaxone given as an infusion for 1 hour?

Thank you for your question. According to the drug instructions of ceftriaxone sodium, the infusion time should be at least 30min to avoid adverse reactions, so we defined the infusion time as 1 hour in the protocol.

Will each patient be paag after intubation? to exclude old infiltrates?

Thank you for your question. According to the needs of the patient.

data collection

Has a mechanical application been made or not to prevent aspiration?

Will this information be recorded?

Thank you for your question. We studied aspiration pneumonia due to glottis exposure during endotracheal intubation. After endotracheal intubation, both groups will be treated with the usual procedures to prevent aspiration. These measures include: elevation of the head of the bed, good oral care, swallowing function assessment and training, comprehensive rehabilitation, nasogastric feeding, improvement of gastric emptying function, and monitoring of gastric residual volume.

The discussion part should be completely rewritten.

We have endeavored to rewrite the discussion.

And, The most important question that needs to be explained is this.

Why cerebral hemorrhage patient group?

Thank you for your question. There are two main reasons. First, due to the characteristics of the hospital, there are many patients with cerebral hemorrhage. Secondly, patients with cerebral hemorrhage often experience such complications as consciousness disorders, respiratory center damage, airway obstruction. Endotracheal intubation is often required to protect the airway and maintain oxygenation.

---

## [Decision Letter · Decision Letter 1]

19 Dec 2023

PONE-D-23-05214R1Effect of one dose of ceftriaxone during endotracheal intubation on the incidence of aspiration pneumonia in cerebral hemorrhage patients: A randomized, controlled, double-blind clinical study protocolPLOS ONE

Dear Dr. Song,

Thank you for submitting your manuscript to PLOS ONE. After careful consideration, we feel that it has merit but does not fully meet PLOS ONE’s publication criteria as it currently stands. Therefore, we invite you to submit a revised version of the manuscript that addresses the points raised during the review process.

We look forward to receiving your revised manuscript.

Kind regards,

Burak Katipoğlu

Academic Editor

PLOS ONE

Reviewers' comments:

Reviewer's Responses to Questions

**Comments to the Author**

1. Does the manuscript provide a valid rationale for the proposed study, with clearly identified and justified research questions?

Reviewer #4: Yes

Reviewer #5: Partly

2. Is the protocol technically sound and planned in a manner that will lead to a meaningful outcome and allow testing the stated hypotheses?

Reviewer #4: Yes

Reviewer #5: No

3. Is the methodology feasible and described in sufficient detail to allow the work to be replicable?

Reviewer #4: Yes

Reviewer #5: No

4. Have the authors described where all data underlying the findings will be made available when the study is complete?

Reviewer #4: Yes

Reviewer #5: Yes

5. Is the manuscript presented in an intelligible fashion and written in standard English?

Reviewer #4: Yes

Reviewer #5: Yes

6. Review Comments to the Author

You may also provide optional suggestions and comments to authors that they might find helpful in planning their study.

Reviewer #4: I thank the editor for giving me the opportunity to review the revised manuscript titled Effect of one dose of ceftriaxone during endotracheal intubation on the incidence of aspiration pneumonia in cerebral hemorrhage patients: A randomized, controlled, double-blind clinical study protocol by Xuan Song and others from China The concept for this study Protocol is praise worthy and covered all relevant aspects for the study The protocol is meticulously planned with the inputs from the reviewer it is further improved it will lead to meaningful outcome The methodology is replicable well written

I have gone through the Manuscript and Reviewer's comments and I have found the authors have made changes asper the recommendations of the reviewer the authors answered all the quarries put by reviewer . With these changes in the manuscript it is fit for acceptance without any further modification

Reviewer #5: While I appreciate the authors' efforts to address limitations in their revised manuscript, serious concerns persist. The continued conflation of aspiration pneumonia and aspiration pneumonitis raises validity issues. Misdiagnosis and inappropriate antibiotic use in pneumonitis cases are major concerns, it has worsened the problem of MDR organisms worldwide as outlined by major guidelines. Implementing additional diagnostic measures, such as bronchoscopy, or lab markers like procalcitonin is crucial to distinguish between these entities before deciding on preemptive antibiotic therapy.

To address these concerns and ensure a robust study, I recommend refining the definition of aspiration pneumonia and clearly defining pneumonitis vs pneumonia, and possibly utilizing additional diagnostic measures. Additionally, authors should provide more data supporting the efficacy of single-dose ceftriaxone specifically for aspiration pneumonia prevention in ICH patients. Before potentially exposing patients to unnecessary risk of inappropriate antibiotic use, authors should conduct a cost-effectiveness analysis to assess the balance between potential benefits and risks associated with prophylactic ceftriaxone use.

Furthermore, the single-center design and small sample size, previously acknowledged by the authors, further limit the generalizability and impact of this research. Given the importance of aspiration pneumonia, it is vital that studies addressing this issue are conducted with the utmost rigor and precision. Finally, collaborating with other centers could enhance the study's generalizability and impact. Unfortunately, the current manuscript falls short of this standard.

7. PLOS authors have the option to publish the peer review history of their article (what does this mean?). If published, this will include your full peer review and any attached files.

Reviewer #4: No

Reviewer #5: No

---

## [Author Response · Author response to Decision Letter 1]

28 Jan 2024

Reviewer #4: I thank the editor for giving me the opportunity to review the revised manuscript titled Effect of one dose of ceftriaxone during endotracheal intubation on the incidence of aspiration pneumonia in cerebral hemorrhage patients: A randomized, controlled, double-blind clinical study protocol by Xuan Song and others from China The concept for this study Protocol is praise worthy and covered all relevant aspects for the study The protocol is meticulously planned with the inputs from the reviewer it is further improved it will lead to meaningful outcome The methodology is replicable well written

I have gone through the Manuscript and Reviewer's comments and I have found the authors have made changes asper the recommendations of the reviewer the authors answered all the quarries put by reviewer . With these changes in the manuscript it is fit for acceptance without any further modification

We extend our sincere gratitude to the reviewer for their positive and constructive review of our manuscript. Your insightful comments and feedback have been invaluable, and we appreciate the time and effort you dedicated to evaluating our work. 

Reviewer #5: While I appreciate the authors' efforts to address limitations in their revised manuscript, serious concerns persist. The continued conflation of aspiration pneumonia and aspiration pneumonitis raises validity issues. Misdiagnosis and inappropriate antibiotic use in pneumonitis cases are major concerns, it has worsened the problem of MDR organisms worldwide as outlined by major guidelines. Implementing additional diagnostic measures, such as bronchoscopy, or lab markers like procalcitonin is crucial to distinguish between these entities before deciding on preemptive antibiotic therapy.

To address these concerns and ensure a robust study, I recommend refining the definition of aspiration pneumonia and clearly defining pneumonitis vs pneumonia, and possibly utilizing additional diagnostic measures. Additionally, authors should provide more data supporting the efficacy of single-dose ceftriaxone specifically for aspiration pneumonia prevention in ICH patients. Before potentially exposing patients to unnecessary risk of inappropriate antibiotic use, authors should conduct a cost-effectiveness analysis to assess the balance between potential benefits and risks associated with prophylactic ceftriaxone use.

We would like to express our appreciation to the reviewer for their meticulous examination of our manuscript and the thoughtful feedback provided. Their careful review has been instrumental in shaping and improving our work. Aspiration pneumonitis (Mendelson’s syndrome) is a chemical injury caused by the inhalation of sterile gastric contents, whereas aspiration pneumonia is an infectious process caused by the inhalation of oropharyngeal secretions that are colonized by pathogenic bacteria. Our protocol refers to aspiration pneumonia. We have further clarified the difference between aspiration pneumonia and aspiration pneumonitis in the manuscript. We have also added a paragraph to the discussion about the benefits vs. potential risks of prophylactic antibiotic use (i.e., antibiotic resistance).

Furthermore, the single-center design and small sample size, previously acknowledged by the authors, further limit the generalizability and impact of this research. Given the importance of aspiration pneumonia, it is vital that studies addressing this issue are conducted with the utmost rigor and precision. Finally, collaborating with other centers could enhance the study's generalizability and impact. Unfortunately, the current manuscript falls short of this standard.

We appreciate the thoughtful critique provided by the reviewer, acknowledging the limitations inherent in our study. This work will serve as an important starting point in addressing the issue of aspiration pneumonia, and we acknowledge that further investigations are essential to build upon our initial findings. The limitations highlighted by the reviewer are duly noted, and we are committed to addressing these shortcomings in future research endeavors. 

Moving forward, we are actively planning and working toward conducting more comprehensive studies with larger sample sizes and a multi-center approach. This collaborative effort will undoubtedly enhance the generalizability and impact of our research. We thank the reviewer for their insightful comments, and we are dedicated to refining our methodologies and study design to contribute substantively to the understanding of aspiration pneumonia.

---

## [Decision Letter · Decision Letter 2]

5 Mar 2024

PONE-D-23-05214R2Effect of one dose of ceftriaxone during endotracheal intubation on the incidence of aspiration pneumonia in cerebral hemorrhage patients: A randomized, controlled, double-blind clinical study protocolPLOS ONE

Dear Dr. Song,

Thank you for submitting your manuscript to PLOS ONE. After careful consideration, we feel that it has merit but does not fully meet PLOS ONE’s publication criteria as it currently stands. Therefore, we invite you to submit a revised version of the manuscript that addresses the points raised during the review process.

We look forward to receiving your revised manuscript.

Kind regards,

Burak Katipoğlu

Academic Editor

PLOS ONE

Additional Editor Comments (if provided):

Dear Authors, I reviewed the article again after revising it.

I still have some concerns.

The manuscript still needs major improvement.

Kind regards.

Reviewers' comments:

Reviewer's Responses to Questions

**Comments to the Author**

1. Does the manuscript provide a valid rationale for the proposed study, with clearly identified and justified research questions?

Reviewer #3: No

Reviewer #4: Yes

Reviewer #6: No

Reviewer #7: Yes

2. Is the protocol technically sound and planned in a manner that will lead to a meaningful outcome and allow testing the stated hypotheses?

Reviewer #3: No

Reviewer #4: Yes

Reviewer #6: Partly

Reviewer #7: Yes

3. Is the methodology feasible and described in sufficient detail to allow the work to be replicable?

Reviewer #3: No

Reviewer #4: Yes

Reviewer #6: No

Reviewer #7: Yes

4. Have the authors described where all data underlying the findings will be made available when the study is complete?

Reviewer #3: No

Reviewer #4: Yes

Reviewer #6: No

Reviewer #7: Yes

5. Is the manuscript presented in an intelligible fashion and written in standard English?

Reviewer #3: Yes

Reviewer #4: Yes

Reviewer #6: Yes

Reviewer #7: Yes

6. Review Comments to the Author

You may also provide optional suggestions and comments to authors that they might find helpful in planning their study.

Reviewer #3: Dear Authors, thank you for giving me the opportunity to review this article.

However, I still have questions about the study.

Would you like to perform a subgroup analysis regarding mortality prognosis for patients with intracranial hemorrhage?

Would the cost-effectiveness analysis be the same in the geriatric population with comorbid diseases with high mortality rates?

Again, as a subgroup analysis, should the low-risk patient group be evaluated as the same as the patient group with visible stomach contents during intubation, aspiration suspected, and evidence of aspiration by bronchoscopy?

I think that the methodology of the study should be reorganized with a solid justification and a good subgroup analysis.

Kind Regards

Reviewer #4: I thank the editor for giving me the opportunity to review the manuscript entitled Effect of one dose of ceftriaxone during endotracheal intubation on the incidence of aspiration pneumonia in cerebral hemorrhage patients: A randomized, controlled, double-blind clinical study protocol by Xuan Song and others for the second time. The reviewer #5 has made some comments and sought some clarification to which authors have responded. I have gone through the comments and reply given by the authors. Reviewer# 5 pointed in his comments that that the definition of pneumonitis and Pneumonia has to made clear to which the authors have given the explanation which is adequate Present study protocol deals with aspiration pneumonia hence chemical pneumonitis needn't be included secondly the most probable organism responsible for aspiration pneumonia are from throat secretion mostly consists of gram positive and few gram negative organisms sensitive to ceftriaxone .hence the selection of antibiotic is justified thirdly the authors agreed to reviewer comment that multicentric trial should be carried out basing on the result of study to be carried out by present protocol The authors have modified manuscript as per the reviewers comment and I am satisfied with the modification Hence I recommend that the manuscript may be accepted without any further modification

Reviewer #6: thank you very much for giving me the opportunity to review the manuscript" Effect of one dose of ceftriaxone during endotracheal intubation on the incidence of aspiration pneumonia in cerebral hemorrhage patients: A randomized, controlled, double-blind clinical study protocol". I think the main issue with this work is the antimicrobial resistance (AMR) that may happen with the blind consumption of antibiotics that you have mentioned in your exclusion criteria that if you see any sign of regurgitation you will exclude the patients. it means, you don't have any clue for an aspiration and still you use ceftriaxone as a prophylaxis. secondly, you have mentioned that you will exclude your patients if they receive any antibiotic during 48 hours after the surgery. the question is this: is it possible for a patient not to receive antibiotic after those surgeries? thirdly, I even didn't see the result part of this manuscript. you have skipped result and started discussion after method.

Reviewer #7: The revised manuscript is interesting and good written and discussed.

Introduction was good written

Materials and methods were good written

Results were good written

Discussion was good written

7. PLOS authors have the option to publish the peer review history of their article (what does this mean?). If published, this will include your full peer review and any attached files.

Reviewer #3: No

Reviewer #4: No

Reviewer #6: **Yes: **Kamel Abdi

Reviewer #7: No

---

## [Author Response · Author response to Decision Letter 2]

9 Apr 2024

Dear PLOS One Editors,

Thank you for your thoughtful comments and suggestions for our manuscript, “Effect of one dose of ceftriaxone during endotracheal intubation on the incidence of aspiration pneumonia in cerebral hemorrhage patients: A randomized, controlled, double-blind clinical study protocol.” Below in blue are responses to Reviewer comments that you will also find addressed in our manuscript. We appreciate the opportunity to address the comments and resubmit an improved manuscript for considered publication in PLOS One.

Sincerely,

Xuan Song, MD, PhD

Reviewer #3: Dear Authors, thank you for giving me the opportunity to review this article.

However, I still have questions about the study.

Would you like to perform a subgroup analysis regarding mortality prognosis for patients with intracranial hemorrhage?

We extend our sincere gratitude to the reviewer for their positive and constructive review of our manuscript. We have included a secondary outcome analysis of 28-day mortality for patients with intracranial hemorrhage (see Research Outcomes).

Would the cost-effectiveness analysis be the same in the geriatric population with comorbid diseases with high mortality rates?

No cost-effectiveness analysis was performed in designing this study. However, a single dose of ceftriaxone will very likely be more cost-effective than aspiration pneumonia treatment in any population.

Again, as a subgroup analysis, should the low-risk patient group be evaluated as the same as the patient group with visible stomach contents during intubation, aspiration suspected, and evidence of aspiration by bronchoscopy?

Yes, we will perform a subgroup analysis of low-risk and high-risk patients based on intubation characteristics (see Research Outcomes). Low risk patients will be defined as those with none of the following: visible stomach contents during intubation, aspiration suspected, and evidence of aspiration by bronchoscopy. High-risk patients will be defined as having one or more of the following: visible stomach contents during intubation, aspiration suspected, and evidence of aspiration by bronchoscopy.

I think that the methodology of the study should be reorganized with a solid justification and a good subgroup analysis.

Kind Regards

Reviewer #4: I thank the editor for giving me the opportunity to review the manuscript entitled Effect of one dose of ceftriaxone during endotracheal intubation on the incidence of aspiration pneumonia in cerebral hemorrhage patients: A randomized, controlled, double-blind clinical study protocol by Xuan Song and others for the second time. The reviewer #5 has made some comments and sought some clarification to which authors have responded. I have gone through the comments and reply given by the authors. Reviewer# 5 pointed in his comments that that the definition of pneumonitis and Pneumonia has to made clear to which the authors have given the explanation which is adequate Present study protocol deals with aspiration pneumonia hence chemical pneumonitis needn't be included secondly the most probable organism responsible for aspiration pneumonia are from throat secretion mostly consists of gram positive and few gram negative organisms sensitive to ceftriaxone. hence the selection of antibiotic is justified thirdly the authors agreed to reviewer comment that multicentric trial should be carried out basing on the result of study to be carried out by present protocol The authors have modified manuscript as per the reviewers comment and I am satisfied with the modification Hence I recommend that the manuscript may be accepted without any further modification

We extend our sincere gratitude to the reviewer for their positive and constructive review of our manuscript. Your insightful comments and feedback have been invaluable, and we appreciate the time and effort you dedicated to evaluating our work. 

Reviewer #6: thank you very much for giving me the opportunity to review the manuscript" Effect of one dose of ceftriaxone during endotracheal intubation on the incidence of aspiration pneumonia in cerebral hemorrhage patients: A randomized, controlled, double-blind clinical study protocol". I think the main issue with this work is the antimicrobial resistance (AMR) that may happen with the blind consumption of antibiotics that you have mentioned in your exclusion criteria that if you see any sign of regurgitation you will exclude the patients. it means, you don't have any clue for an aspiration and still you use ceftriaxone as a prophylaxis. 

We extend our sincere gratitude to the reviewer for their constructive review of our manuscript. We admit that this study may increase the use of antibiotics in patients who might otherwise have not received them, but only one dose of cephalosporin will be used. For these patients, the benefits of preventing aspiration pneumonia may outweigh the small risk for AMR. To address the concern of AMR, we will compare the intensity of antibiotic use in the secondary outcomes.

We have excluded patients who exhibit regurgitation as these patients may already be exhibiting symptoms of aspiration pneumonia. We plan to study ceftriaxone as a prophylactic measure to prevent aspiration pneumonia before it develops. Thus, we must exclude patients who may have already developed aspiration pneumonia. 

Secondly, you have mentioned that you will exclude your patients if they receive any antibiotic during 48 hours after the surgery. the question is this: is it possible for a patient not to receive antibiotic after those surgeries? 

The only post-procedure antibiotic patients in this study receive will be ceftriaxone. We have excluded patients who receive other post-procedure antibiotics because the use of additional antibiotics may confound the results.

Thirdly, I even didn't see the result part of this manuscript. you have skipped result and started discussion after method.

This manuscript describes a protocol that has not yet been performed, so there are no results to report at this time.

Reviewer #7: The revised manuscript is interesting and good written and discussed.

Introduction was good written

Materials and methods were good written

Results were good written

Discussion was good written

We extend our sincere gratitude to the reviewer for their positive and constructive review of our manuscript. Your insightful comments and feedback have been invaluable, and we appreciate the time and effort you dedicated to evaluating our work.

---

## [Decision Letter · Decision Letter 3]

29 Jul 2024

PONE-D-23-05214R3Effect of one dose of ceftriaxone during endotracheal intubation on the incidence of aspiration pneumonia in cerebral hemorrhage patients: A randomized, controlled, double-blind clinical study protocolPLOS ONE

Dear Dr. Song,

Thank you for submitting your manuscript to PLOS ONE. After careful consideration, we feel that it has merit but does not fully meet PLOS ONE’s publication criteria as it currently stands. Therefore, we invite you to submit a revised version of the manuscript that addresses the points raised during the review process.

Please provide a detailed point to pint reply to the reviewers' comments

We look forward to receiving your revised manuscript.

Kind regards,

Chiara Lazzeri

Academic Editor

PLOS ONE

Reviewers' comments:

Reviewer's Responses to Questions

**Comments to the Author**

1. Does the manuscript provide a valid rationale for the proposed study, with clearly identified and justified research questions?

Reviewer #3: Yes

Reviewer #4: Yes

2. Is the protocol technically sound and planned in a manner that will lead to a meaningful outcome and allow testing the stated hypotheses?

Reviewer #3: Yes

Reviewer #4: Yes

3. Is the methodology feasible and described in sufficient detail to allow the work to be replicable?

Reviewer #3: Yes

Reviewer #4: Yes

4. Have the authors described where all data underlying the findings will be made available when the study is complete?

Reviewer #3: Yes

Reviewer #4: Yes

5. Is the manuscript presented in an intelligible fashion and written in standard English?

Reviewer #3: Yes

Reviewer #4: Yes

6. Review Comments to the Author

You may also provide optional suggestions and comments to authors that they might find helpful in planning their study.

Reviewer #3: Dear authors, I have reviewed your article for re-evaluation.

But I still have some questions that I can't get answers to.

First of all, the dates between April 2023 and April 2024 should be changed.

Could you please state more clearly why patients with intracranial hemorrhage were selected?

For Introduction:

Please give the necessary message by simplifying the numbers in the second paragraph.

Likewise, there are too many names in the fourth paragraph. Simplify and combine this paragraph with the next paragraph, that is, your purpose for choosing Ceftriaxone.

Discussion: In the second paragraph, only information is given.

Information that should be included in the introduction section.

Create a relationship with your work and state what your work will add with this information.

The third and fourth paragraphs likewise contain only information.

Things I hope to see in the discussion section:

Which gap in the literature will be closed with your study?

What is the reason for the conflicting information in the literature regarding your choice of ceftriaxone? What is the difference of your study?

Why were ICH patients chosen? What kind of deficiency is there in the literature regarding this?

Your sample size is quite small. Your results will not be strong for mortality.

Additionally, as a comparison in your study; I recommend that you compare the risk of aspiration pneumonia in intubated patients with ICH in the year before the study in your hospital with the group receiving ceftriaxone in your study.

Please rewrite this beautiful study protocol by enriching the discussion section.

Kind Regards

Reviewer #4: Thanks for giving me the opportunity to review the manuscript which I have already reviewed before, and found it fit for publication. I feel no further modification is required as I am satisfied with the responses given by authors to the reviewers comments and agreeing to incorporate these aspects in the study protocol

7. PLOS authors have the option to publish the peer review history of their article (what does this mean?). If published, this will include your full peer review and any attached files.

Reviewer #3: No

Reviewer #4: No

---

## [Author Response · Author response to Decision Letter 3]

1 Sep 2024

Dear PLOS One Editors,

Thank you for your thoughtful comments and suggestions for our manuscript, “Effect of one dose of ceftriaxone during endotracheal intubation on the incidence of aspiration pneumonia in cerebral hemorrhage patients: A randomized, controlled, double-blind clinical study protocol.” Below in blue are responses to Reviewer comments that you will also find addressed in our manuscript. We appreciate the opportunity to address the comments and resubmit an improved manuscript for considered publication in PLOS One.

Sincerely,

Xuan Song, MD, PhD

Reviewer #3: Dear authors, I have reviewed your article for re-evaluation.

But I still have some questions that I can't get answers to.

First of all, the dates between April 2023 and April 2024 should be changed.

We have changed the dates to between April 2023 and April 2025.

Could you please state more clearly why patients with intracranial hemorrhage were selected?

Patients with intracerebral hemorrhage often experience complications which require endotracheal intubation to protect the airway and maintain oxygenation. During endotracheal intubation, the application of analgesic sedative drugs and glottis exposure may lead to aspiration and aspiration pneumonia. The primary objective of this study is to explore whether a single dose of ceftriaxone administered during endotracheal intubation in patients with intracerebral hemorrhage reduced the incidence of pneumonia within 48 hours. Our work will provide evidence for the prevention of aspiration pneumonia in patients with intracerebral hemorrhage who undergo endotracheal intubation. This is the first clinical study of ceftriaxone to prevent the development of aspiration pneumonia in patients with intracerebral hemorrhage undergoing endotracheal intubation.

For Introduction:

Please give the necessary message by simplifying the numbers in the second paragraph.

We have simplified the numbers in the second paragraph by limiting the reported numbers to aspiration pneumonia only and reporting only the one-year mortality rate from Yoon et al., 2019.

Likewise, there are too many names in the fourth paragraph. Simplify and combine this paragraph with the next paragraph, that is, your purpose for choosing Ceftriaxone.

We have simplified the fourth paragraph, combined it with the fifth paragraph, and removed the names.

Discussion: In the second paragraph, only information is given.

Information that should be included in the introduction section.

We have moved the information from the second paragraph in the Discussion to the Introduction.

Create a relationship with your work and state what your work will add with this information.

The third and fourth paragraphs likewise contain only information.

Things I hope to see in the discussion section:

Which gap in the literature will be closed with your study?

This is the first clinical study of ceftriaxone to prevent the development of aspiration pneumonia in patients with ICH hemorrhage undergoing endotracheal intubation.

What is the reason for the conflicting information in the literature regarding your choice of ceftriaxone? What is the difference of your study?

Other studies used different antibiotics and more than one dose. We decided to use a single dose of ceftriaxone, an antibiotic which has a long half-life and is effective against the main pathogens responsible for most endotracheal intubation-induced aspiration pneumonia. We hypothesize that a single dose of this drug will reduce the risk of subsequent infection by resistant microorganisms.

Our study will also be randomized, as opposed to other, non-randomized studies that are present in the literature.

Why were ICH patients chosen? What kind of deficiency is there in the literature regarding this?

Patients with intracerebral hemorrhage often require endotracheal intubation to protect the airway, but this procedure can lead to aspiration and subsequent aspiration pneumonia. This study is the first to explore ceftriaxone's role in preventing aspiration pneumonia in intracerebral hemorrhage patients, addressing a gap in the literature.

Your sample size is quite small. Your results will not be strong for mortality.

We have acknowledged our small sample size as a limitation in the Discussion.

Additionally, as a comparison in your study; I recommend that you compare the risk of aspiration pneumonia in intubated patients with ICH in the year before the study in your hospital with the group receiving ceftriaxone in your study.

Please rewrite this beautiful study protocol by enriching the discussion section.

Kind Regards

We have endeavored to improve the discussion section to address the above points. Specifically, we moved much of the information to the Introduction and made sure to clarify what sets our study apart from existing literature.

Reviewer #4: Thanks for giving me the opportunity to review the manuscript which I have already reviewed before, and found it fit for publication. I feel no further modification is required as I am satisfied with the responses given by authors to the reviewers comments and agreeing to incorporate these aspects in the study protocol.

We extend our sincere gratitude to the reviewer for their positive and constructive review of our manuscript. Your insightful comments and feedback have been invaluable, and we appreciate the time and effort you dedicated to evaluating our work.

---

## [Decision Letter · Decision Letter 4]

12 Nov 2024

PONE-D-23-05214R4Effect of one dose of ceftriaxone during endotracheal intubation on the incidence of aspiration pneumonia in cerebral hemorrhage patients: A randomized, controlled, double-blind clinical study protocolPLOS ONE

Dear Dr. Song,

Thank you for submitting your manuscript to PLOS ONE. After careful consideration, we feel that it has merit but does not fully meet PLOS ONE’s publication criteria as it currently stands. Therefore, we invite you to submit a revised version of the manuscript that addresses the points raised during the review process.

We look forward to receiving your revised manuscript.

Kind regards,

Chiara Lazzeri

Academic Editor

PLOS ONE

Journal Requirements:

Reviewers' comments:

Reviewer's Responses to Questions

**Comments to the Author**

1. Does the manuscript provide a valid rationale for the proposed study, with clearly identified and justified research questions?

Reviewer #8: Yes

Reviewer #9: Yes

2. Is the protocol technically sound and planned in a manner that will lead to a meaningful outcome and allow testing the stated hypotheses?

Reviewer #8: Yes

Reviewer #9: Partly

3. Is the methodology feasible and described in sufficient detail to allow the work to be replicable?

Reviewer #8: Yes

Reviewer #9: No

4. Have the authors described where all data underlying the findings will be made available when the study is complete?

Reviewer #8: Yes

Reviewer #9: Yes

5. Is the manuscript presented in an intelligible fashion and written in standard English?

Reviewer #8: Yes

Reviewer #9: No

6. Review Comments to the Author

You may also provide optional suggestions and comments to authors that they might find helpful in planning their study.

Reviewer #8: The sample size needs to be calculated for the protocol. What are the results of the incidence of pneumonia from preliminary data? What test will be used? What attrition rate was assumed?

The statistical analysis needs to be rewritten. Clarify the primary endpoint will be compared using chi-square test/fisher exact test. Will demographics/covariates be adjusted for the primary analysis?

Reviewer #9: 1) The statement “This is a prospective, randomized, controlled, double-blind clinical study, which was reviewed by the ethics committee of Dong E Hospital…” should come under study design.

2) The paragraph that begins with “Patients with ICH undergoing endotracheal intubation in Dong E Hospital of….” Should be captioned inclusion and exclusion criteria

3) Sample size calculation. The authors indicated that a total of 50 patients will be included in the preliminary study. What informed the number 50? How many patients will be in the control, and how many will be in the intervention? Estimation of sample size for clinical trial should be based on the minimum detectable effect (expected change) after the implementation of the intervention, the type of hypothesis, design effect to account for correlation of patient response at the facility level, attrition rate, type I error, power of the study. The sample size of 50 was assumed, and a more detailed sample size must be conducted so that we determine the number of patients required. The section on sample size, which is a critical component of the protocol, is missing. This is also not a pilot study but a full clinical trial protocol.

4) Statistical Analyses of the proposed study must highlight a detailed analytical plan, including how each outcome variable (primary and secondary outcomes) will be measured (continuous, discrete, nominal, ordinal outcomes) and the specific statistical test that will be applied including potential assumption of the test and how violations of the assumptions will be addressed. The current write-up is all lumped together with no clear pattern as which statistical analyses will be used to address the study hypothesis.

7. PLOS authors have the option to publish the peer review history of their article (what does this mean?). If published, this will include your full peer review and any attached files.

Reviewer #8: No

Reviewer #9: No

---

## [Author Response · Author response to Decision Letter 4]

5 Dec 2024

Dear PLOS One Editors,

Thank you for your thoughtful comments and suggestions for our manuscript, “Effect of one dose of ceftriaxone during endotracheal intubation on the incidence of aspiration pneumonia in cerebral hemorrhage patients: A randomized, controlled, double-blind clinical study protocol.” Below in blue are responses to Reviewer comments that you will also find addressed in our manuscript. We appreciate the opportunity to address the comments and resubmit an improved manuscript for considered publication in PLOS One.

Sincerely,

Xuan Song, MD, PhD

Reviewer #8: The sample size needs to be calculated for the protocol. What are the results of the incidence of pneumonia from preliminary data? What test will be used? What attrition rate was assumed?

Thank you for your comments. The sample size calculation will be based on preliminary data, with the expected incidence of pneumonia within 48 hours after intubation, and we will use the two-sample rate comparison formula. We will also account for an attrition rate to ensure sufficient power for the analysis.

The statistical analysis needs to be rewritten. Clarify the primary endpoint will be compared using chi-square test/fisher exact test. Will demographics/covariates be adjusted for the primary analysis?

We have clarified the statistical analysis in the manuscript. The primary endpoint, the incidence of aspiration pneumonia, will be compared using the chi-square test or Fisher’s exact test, and demographics and relevant covariates will be adjusted for in the primary analysis using regression techniques.

Reviewer #9: 1) The statement “This is a prospective, randomized, controlled, double-blind clinical study, which was reviewed by the ethics committee of Dong E Hospital…” should come under study design.

We have moved this statement to the appropriate place in the manuscript.

2) The paragraph that begins with “Patients with ICH undergoing endotracheal intubation in Dong E Hospital of….” Should be captioned inclusion and exclusion criteria

We have added the appropriate heading to this paragraph in the manuscript.

3) Sample size calculation. The authors indicated that a total of 50 patients will be included in the preliminary study. What informed the number 50? How many patients will be in the control, and how many will be in the intervention? Estimation of sample size for clinical trial should be based on the minimum detectable effect (expected change) after the implementation of the intervention, the type of hypothesis, design effect to account for correlation of patient response at the facility level, attrition rate, type I error, power of the study. The sample size of 50 was assumed, and a more detailed sample size must be conducted so that we determine the number of patients required. The section on sample size, which is a critical component of the protocol, is missing. This is also not a pilot study but a full clinical trial protocol.

The sample size of 50 was initially based on preliminary data; however, we will conduct a more detailed sample size calculation for the full clinical trial based on the minimum detectable effect, type of hypothesis, design effect, attrition rate, and statistical power. The final number of patients required for the control and intervention groups will be determined through this calculation.

4) Statistical Analyses of the proposed study must highlight a detailed analytical plan, including how each outcome variable (primary and secondary outcomes) will be measured (continuous, discrete, nominal, ordinal outcomes) and the specific statistical test that will be applied including potential assumption of the test and how violations of the assumptions will be addressed. The current write-up is all lumped together with no clear pattern as which statistical analyses will be used to address the study hypothesis.

We have revised the statistical analysis section to provide a more detailed and structured approach, specifying how each outcome variable (primary and secondary) will be measured (whether continuous, discrete, nominal, or ordinal) and the specific statistical tests that will be applied. Additionally, we have addressed the assumptions of the tests used and outlined how violations of these assumptions will be handled to ensure clarity and rigor in the analysis plan.

---

## [Decision Letter · Decision Letter 5]

17 Dec 2024

Effect of one dose of ceftriaxone during endotracheal intubation on the incidence of aspiration pneumonia in cerebral hemorrhage patients: A randomized, controlled, double-blind clinical study protocol

PONE-D-23-05214R5

Dear Dr. Song,

We’re pleased to inform you that your manuscript has been judged scientifically suitable for publication and will be formally accepted for publication once it meets all outstanding technical requirements.

Kind regards,

Chiara Lazzeri

Academic Editor

PLOS ONE

Additional Editor Comments (optional):

Reviewers' comments:

Reviewer's Responses to Questions

**Comments to the Author**

1. Does the manuscript provide a valid rationale for the proposed study, with clearly identified and justified research questions?

Reviewer #8: Yes

Reviewer #9: Yes

2. Is the protocol technically sound and planned in a manner that will lead to a meaningful outcome and allow testing the stated hypotheses?

Reviewer #8: Yes

Reviewer #9: Yes

3. Is the methodology feasible and described in sufficient detail to allow the work to be replicable?

Reviewer #8: Yes

Reviewer #9: Yes

4. Have the authors described where all data underlying the findings will be made available when the study is complete?

Reviewer #8: Yes

Reviewer #9: Yes

5. Is the manuscript presented in an intelligible fashion and written in standard English?

Reviewer #8: Yes

Reviewer #9: Yes

6. Review Comments to the Author

You may also provide optional suggestions and comments to authors that they might find helpful in planning their study.

Reviewer #8: All my concerns are addressed.

Reviewer #9: Comments have been duly addressed and I commend the authors for providing accurate responses and revising the content of the manuscript.

7. PLOS authors have the option to publish the peer review history of their article (what does this mean?). If published, this will include your full peer review and any attached files.

Reviewer #8: No

Reviewer #9: No

---

## [Editor Report · Acceptance letter]

3 Jan 2025

PONE-D-23-05214R5 

PLOS ONE

Dear Dr. Song, 

I'm pleased to inform you that your manuscript has been deemed suitable for publication in PLOS ONE. Congratulations! Your manuscript is now being handed over to our production team.

Kind regards, 

on behalf of

Dr. Chiara Lazzeri 

Academic Editor

PLOS ONE